# Field Performance of Disease-Free Plants of Ginger Produced by Tissue Culture and Agronomic, Cytological, and Molecular Characterization of the Morphological Variants

Xiaoqin Zhao [1,2], Shuangying Yu [1], Yida Wang [1], Dongzhu Jiang [1], Yiming Zhang [1], Liu Hu [1], Yongxing Zhu [1], Qie Jia [1], Junliang Yin [3], Yiqing Liu [1,*] and Xiaodong Cai [1,*]

1   Spice Crops Research Institute, College of Horticulture and Gardening, Yangtze University, Jingzhou 434025, China
2   Jingzhou Jiazhiyuan Biotechnology Co., Ltd., Jingzhou 434025, China
3   College of Agriculture, Yangtze University, Jingzhou 434025, China
*   Correspondence: liung906@163.com (Y.L.); caixiao.dong@163.com (X.C.)

**Abstract:** Ginger (*Zingiber officinale* Rosc.) is an important spice crop valued for its flavored and medical properties. It is susceptible to soil-borne diseases, which can cause considerable economic loss to growers. In vitro culture is feasible for the propagation of disease-free ginger plants, but has several disadvantages when producing seed rhizomes that can be commercially used, such as long cultivation cycles (usually 2–3 years) and occurrence of somaclonal variation. In this study, dynamic changes in the morphological characteristics of in vitro-propagated disease-free plants of 'Wuling' ginger were evaluated by continuous observation and measurement at 30-day intervals, and morphological variants were screened and characterized by agronomic, cytological, and molecular analysis at harvest. Results showed that the plants grew rapidly within 120 days after planting, and the most active growth period was from 60 to 120 days. Eight plants with clear and stable morphological differences were screened out from approximately 2000 plants grown in the field, and they could be classified into two groups (VT1 and VT2) based on tiller number, plant height, leaf color, and leaf shape. By flow cytometry analysis and chromosome counting, the VT1 was confirmed to be diploid, with the shortest plant height, the largest number of tillers and rhizome knobs, and the smallest tiller diameter and rhizome size among the three types of plants. The VT2 was mixoploid, consisting of diploid and tetraploid cells, with significantly reduced tiller number and rhizome knobs, significantly larger stomatal guard cells/apertures, and significantly lower stomatal density. SSR analysis detected DNA band profile changes in six out of the eight variants, including one plant of the VT1 and all the VT2 plants. The findings of this study might contribute to the commercial production of disease-free seed rhizomes in ginger, and the characterized somaclonal variants could provide useful germplasm resources for future breeding.

**Keywords:** chromosome counting; flow cytometry analysis; growth dynamics; mixoploid; somaclonal variation; SSR analysis

## 1. Introduction

Ginger (*Zingiber officinale* Rosc.) is an important perennial herbaceous spice crop with diverse culinary properties and medicinal potential cultivated commercially worldwide [1–3]. It is commonly served as traditional condiments due to its distinct aromatic flavor and abundant nutritive components [2,3]. In addition to its culinary uses, ginger also possesses diverse medicinal functions and health-beneficial properties [4,5]. In traditional Chinese medicine, dried or fresh ginger rhizomes are commonly used as medicinal ingredients for preventing or relieving symptoms such as the common cold, headache, nausea, and vomiting [5,6]. Many studies have confirmed that ginger extracts contain a

variety of bioactive compounds with antioxidant, antimicrobial, anti-inflammatory, anti-tumor, and hypoglycemic and hypolipidemic effects [2–5]. More recently, ginger extracts were reported to have considerable inhibiting effects against plant pathogens such as *Phytophthora colocasiae* [7] and pests such as *Melanaphis sorghi* [8].

During the cultivation and storage of ginger, diseases caused by various microbial pathogens or nematodes are among the major problems faced by growers, which can largely reduce the crop yield and quality [9–11]. Ginger crops are susceptible to several types of soil-borne diseases, which can cause considerable economic loss [10,12], such as bacterial wilt disease caused by *Ralstonia solanacearum* [10], rhizome rot caused by *Fusarium oxysporum* [13], and nematode disease caused by *Meloidogyne* spp. [12]. In the absence of sexual reproduction, ginger is generally vegetatively propagated by the division of mature rhizomes known as seed rhizomes. As a consequence, this conventional propagation approach is threatened by disease transmission through the mother rhizomes infected by these pathogens after continuous intensive cultivation. Crop productivity and quality thereby can be tremendously reduced when the rhizomes with diseases are used as seed rhizomes [14]. Accordingly, high efficiency in the production of disease-free seed rhizomes is an imperative way to minimize the infection rate and increase the economic benefits of ginger.

Plant tissue culture techniques have been extensively used in many crops to produce genetically identical and disease-free plants [15]. In ginger, well-developed plants can be regenerated in vitro from active buds forced from rhizomes (approximately 0.3–0.5 cm in height), and the regenerants were proved to be disease free based on disease indexing or diagnostic tests during the growing season and rhizome storage [16,17]. Additionally, in vitro culture combined with micro-rhizome induction could be an alternative to eliminate the rhizome-borne pathogens in ginger [18]. Therefore, in vitro culture is a feasible approach for the mass propagation of disease-free plants with a high multiplication rate in ginger.

Before commercial production, the field performance of in vitro-regenerated plants is an important aspect that needs further attention. Generally, tissue culture-derived plants are expected to have several advantages over traditionally propagated plants, such as higher growth vigor, increased economic yield, improved crop quality, and reduced disease infection rate [19,20]. However, earlier studies showed that micropropagated plants had a much lower yield with smaller knobs compared with rhizome-derived plants in the first growing season in ginger [21,22]. It will take at least 2 to 3 growing seasons to produce an acceptable yield of seed rhizomes with an adequate size of knobs that can be used commercially [23], which will cause increased costs of the production of disease-free ginger rhizomes and thus hindered their commercial application. The yield of ginger was reported to be highly correlated with its morphological traits such as tiller diameter, tiller number and plant height [22–24]. Therefore, it is feasible to increase the productivity of disease-free ginger plants by improving the morphological traits in the early growing season through good field management. Insights into dynamic changes in morphological parameters are favorable to adopt suitable field management. However, few reports are available on detailed evaluations of field performance of disease-free plants of ginger produced by tissue culture [23].

In addition, somaclonal variation including genetic and epigenetic alterations can occur spontaneously in a variety of plant species during tissue culture, as can be ascertained by several strategies comprising morphological, cytological, biochemical, and molecular methods [25,26]. The occurrence of somaclonal variation is an unfavorable aspect when producing true-to-type plants by tissue culture [26]. However, it can create various phenotypes and increase genetic variability for breeders. Presently, tissue culture-induced somaclonal variation has become important sources of variability for crop genetic improvement, especially for those asexually reproduced plant species [26,27].

Conventional sexual hybridization is limited in ginger for its poor flowering, pollen viability, and seed set, resulting in its narrow genetic variability [28]. As a result, biotechnological approaches such as mutation breeding [29], polyploidy breeding [30], genetic trans-

formation [31], and somaclonal variation [32,33] have been attempted in ginger to develop new cultivars. For the somaclonal variants of ginger, great differences has been observed in their agronomic traits, nutritional components [32], and disease resistance [33]. In addition, somaclonal variation in ginger was also identified at the DNA level. For example, Abd El-Hameid et al. [34] (2020) found that somaclonal variation occurred in callus-regenerated plants of ginger by ISSR analysis, and speculated that 2,4-dichlorophenoxyacetic acid (2,4-D) in the callus induction medium might be responsible for the genetic changes. Although the occurrence of somaclonal variation is an adverse factor in the commercial production of disease-free seed rhizomes in ginger, the screened and characterized somaclonal variants are valuable materials to enrich its genetic diversity.

'Wuling' ginger, a well-known local ginger cultivar in Wuling Mountain area, China, is characterized by a strong spicy flavor but a relatively low yield. In this study, Morphological parameters of the in vitro produced disease-free plants of 'Wuling' ginger were measured and analyzed during the whole growing season, and morphological variation plants were observed closely and subjected to agronomic, cytological, and molecular evaluation at harvest. The purposes of this study were to understand the field performance of the disease-free plants and screen promising somaclonal lines for genetic improvement in ginger.

## 2. Materials and Methods

### 2.1. Preparation of Plant Materials

'Wuling' ginger is a local ginger cultivar widely grown in Wuling Mountain area, China for its strong spicy flavor. In our previous work, tissue-cultured plants of 'Wuling' ginger were obtained by shoot tip culture and were proved to be disease-free by RT-PCR test (data not published). Prior to the field experiment, the plants had been maintained on MS basal medium containing 1 mg·L$^{-1}$ 6-BA, 0.2 mg·L$^{-1}$ IBA, 3% (*w/v*) sucrose, and 0.8% (*w/v*) agar (pH 5.8) for approximately three years with a subculture interval of two months. For the mass production of the plants to ensure adequate materials for further experiment, the maintained plants were cut in a biological safety cabinet into shoot clusters of approximately 1 cm in height, and cultured on the above medium at 25 ± 1 °C and 14 h/10 h (light/dark) cycles under cool white fluorescent lamps of approximately 55 μmol m$^{-2}$ s$^{-1}$. For acclimation, well-rooted plants were removed from the culture vessels and washed carefully under running tap water to remove any agar residues. Thereafter, the plants of a similar height (approximately 6 cm) were transplanted individually to seedling trays (6 cm × 6 cm × 11 cm/cell) filled with culture substrates comprising of sphagnum peat soil (Pindstrup Mosebrug A/S, Pindstrup, Denmark) and perlite at 3:1 (*v/v*) ratio. Then, the trays were covered with plastic films to maintain a suitable humidity and incubated in a greenhouse at 28/20 °C with a photoperiod of 14 h light/10 h dark) and a light intensity level of approximately 130 μmol m$^{-2}$ s$^{-1}$. The plastic films were removed two weeks later, and the plants were cultured for another two weeks without covers to adapt to natural conditions.

### 2.2. Field Planting of the Acclimatized Plants

Once acclimatized, the hardened plants were planted in soil in a simple plastic greenhouse without additional heat or light supply in the experimental field of Yangtze University, Hubei, China (latitude: 30°21′ N, longitude: 112°09′ E, and altitude: 28 m) from late May to mid-December in 2021 during the growing season. The annual average temperature of the site ranges from 15.9 to 16.5 °C, and the annual mean rainfall is approximately 1100–1150 mm. A total of approximately 2000 plants were transferred to the field and planted at a spacing of 20 cm × 65 cm. The soil texture was sandy loam with a pH of 7.3, and the content of the organic matter, alkali hydrolysable nitrogen, available phosphorus, and available potassium in the cultivated layer were 19.8 g·kg$^{-1}$, 65.3 mg·kg$^{-1}$, 32.5 mg·kg$^{-1}$ and 262.2 mg·kg$^{-1}$, respectively.

*2.3. Morphological Measurements, Visual Screening of Morphological Variants, and Main Agronomic Traits Analysis*

Morphological parameters including plant height, tiller number per plant, tiller diameter, leaf number per tiller, and leaf length and width were measured non-destructively at an interval of 30 days after planting. On days 0, 30, 60, 90, and 120 after planting, related data were collected according to the phenotypic descriptors as suggested by previous report [24] using a five-point sampling method, and 20 plants were selected randomly at each point. At least three mature leaves and three mature tillers per plant were used for analysis. The absolute growth rate (AGR) was calculated for all variables at consecutive harvest intervals: AGR = $(M_2 - M_1)/(t_2 - t_1)$, where $M_1$ and $M_2$ were the measured values at times $t_1$ and $t_2$, respectively.

After 90 days of growth in the field, visual screening of morphological variation was commenced, and any plants exhibiting clear morphological differences were labeled. Data were collected on morphological and yield traits at harvest (mid-December, 2021). The chlorophyll content was determined by the SPAD index, as was measured in the second fully expanded leaf from the apex with a SPAD-502 plus portable chlorophyll meter (Konica Minolta, Tokyo, Japan). After the measurement of the aboveground parameters, the plants were pulled out carefully from the ground, washed gently under running water to remove adhering soil, and then dried naturally at the natural temperature in the air. Afterward, main yield traits including rhizome length and height, number of rhizome knobs, and rhizome fresh weight were measured.

*2.4. Stomatal Measurements*

The aperture length/width, stomatal length/width, and stomatal density were measured following the protocol of Chen et al. [35]. Briefly, nail polish imprints obtained from the lower epidermis of the fully expanded leaves were photographed under a Nikon Eclipse Ni-U microscope equipped with a Nikon DS-Ri2 camera (Nikon, Tokyo, Japan). The stomatal characteristics were determined by counting 20 randomly selected microscopic field areas from three leaf samples of each plant.

*2.5. Flow Cytometry Analysis*

Relative nuclear DNA content was quantified using a Beckman Coulter CytoFLEX flow cytometer (Suzhou, China) with a 488 nm argon laser. Nuclei were released by chopping approximately 1.0 cm$^2$ of fresh young leaves using a sharp razor blade in a plastic Petri dish containing 600 μL of ice-cold lysis buffer (0.3% Triton X-100, 10 Mm $MgSO_4 \cdot 7H_2O$, 50 mM KCl, 5 mM HEPES, 2% PVP), and then filtered through a 40 μm nylon mesh filter into a 5 mL sample tube. Thereafter, 10 μg RNase A and 5 μL propidium iodide (PI) were added to the nuclear suspension and incubated in dark for 30 min at room temperature. The suspension was shaken gently for 10 s before sample analysis. For each plant sample, at least three flow cytometry runs were conducted, and in each run a minimum of 3000 particles were recorded.

*2.6. Chromosome Counting*

Chromosome observation was performed using root tip squashes according to the method described by Chen et al. [35] with minor modifications. Prior to chromosome preparation, the harvested rhizomes of the normal-type plants and the variants were placed in sterilized moist sands and maintained in the dark at approximately 25 °C for rooting. After approximately 30 days of incubation, vigorously growing root tips (approximately 1.5 cm in length) were collected in the morning hours and immediately placed in 2 mM 8-hydroxyquinoline solution in darkness for 3.5–4.0 h. After rinsing with tap water for 5 min, the pretreated root tips were fixed in fresh Carnoy's solution (3 absolute ethanol:1 glacial acetic acid, *v/v*) at 4 °C overnight. For slide preparations, the materials were washed with distilled water and hydrolyzed with 1 N HCl at 60 °C for 5 min. Following rinsing with deionized water three times, the softened root tips were transferred onto a glass

slide, and meristematic tissue was isolated from the root tips by using a sharp scalpel and then stained in carbol-fuchsin solution (Solarbio, Beijing, China) for 10 min. Finally, the meristematic tissue was covered with a cover slide and gently pressed to spread the stained cells, and well-spread metaphase plates were observed and photographed under the Nikon microscope at 1000 times magnification.

### 2.7. Genomic DNA Extraction and SSR Analysis

Two randomly selected normal-type plants and the eight screened morphological variants were subjected to SSR analysis for evaluation of the genetic homogeneity. Total DNA was extracted from approximately 100 mg of fresh and young leaves following the modified cetyltrimethylammonium bromide (CTAB) method [36]. The quantity and purity of the isolated DNA were checked by electrophoresis with 0.8% agarose gel at a constant voltage of 100 V for 30 min and a NanoDrop One C spectrophotometer (Thermo Fisher Scientific, Madison, WI, USA) at 260/280 nm. Then, the sample DNA was diluted to 25 ng/µL for SSR analysis.

SSR primers were designed and synthesized based on the transcriptomic data of several local cultivars of ginger in China (unpublished data), and 20 primer sets (Table 1) were selected randomly and used in the SSR analysis. A 20 µL PCR reaction mixture was prepared as follows: 2 µL of temple DNA (25 ng/µL), 10 µL GenStar (Beijing, China) 2 × Taq Master Mix for polyacrylamide gel electrophoresis (PAGE), 1.0 µL of each reverse and forward primer (10 µM), and 6.0 µL ddH$_2$O. SSR-PCR amplification was performed in a T100 thermocycler (BIO-RAD, Singapore) using a touch-down PCR program as follows: The initial denaturation of the DNA at 94 °C for 2 min, followed by 15 cycles of amplification, consisting of denaturation at 94 °C for 30 s, primer firstly annealing at 63 °C for 30 s with 1 °C reduction per cycle, and extension at 72 °C for 30 s, then 25 cycles, consisting of denaturation at 94 °C for 30 s, primer annealing at 50 °C for 30 s, primer extension at 72 °C for 30 s, and a final extension at 72 °C for 10 min. Amplified products were separated by non-denaturing 8% acrylamide gels according to the protocol described by Yu et al. [27].

**Table 1.** Code and sequence of 20 pairs of SSR primers used in this study.

| Code | Forward Primer | Reverse Primer | Product Size (bp) |
|---|---|---|---|
| Ginger02 | CTTCCTTATGTGCGTTTGTGC | TATCTGGAATGTTGATGAAGTTACC | 405 |
| Ginger07 | ATTGGTTGCGGAATAAAGGTGT | AGCAAAATGGATTAAACATTTGGTC | 236 |
| Ginger11 | AAATGGAGAAGGGGAACTAAT | GCTGAATCATCAATCTTTGTAGTTT | 275 |
| Ginger18 | TGGGTATGTATTGAATGGTGTAGGA | ATCCACCAACTGCTGCTGC | 302 |
| Ginger22 | AAAAGGCGTGGGTGCTACAT | TCGAACACCGGACAACAGAG | 282 |
| Ginger25 | ACTCCAGCAGAACCACAACG | TGGAGGTATCCTCGGTGTCC | 455 |
| Ginger31 | CCGACTCGACCAAGGTAAGC | TCCATTGCAGGTGCCTCTACT | 412 |
| Ginger37 | ATTTCTAGCCAGTTTAGCTGAGCAA | CGTGGAGGAGCCACTTCCG | 283 |
| Ginger44 | CTGAAGCAGATTGTTGTGGTCG | GCGCGACTGACACGACAGG | 324 |
| Ginger49 | CCTCTGTTCAGTTGTTGCCTTGC | ACCGCACGGGCTGTGGATA | 328 |
| Ginger52 | TGGTTCAGTATTTCGGTGGT | ACCAGACTGAGAAGTGGCTAA | 248 |
| Ginger58 | GTCTGTACCATCGGGTTTTGTTA | TTATACCTTGAGATAGGGATGCC | 361 |
| Ginger60 | AGCGACTATTGTGCTTGGGTT | AGGGTGCTCAATAATGACAGA | 285 |
| Ginger69 | CTTCCTTATGTGCGTTTGTGC | TATCTGGAATGTTGATGAAGTTACC | 405 |
| Ginger73 | AGATCAACATAAATGATCTGGTGGC | CGCAAGCCAAGCAAACAAGG | 187 |
| Ginger77 | GATTTACTTTCAACCAGTCAACCCTT | CACTTGCATCACTCTGATCAACA | 322 |
| Ginger82 | ATGGGAGACTCAGGTGGTGT | ACCAACAAATGGAGGAAGAG | 266 |
| Ginger84 | ACTGCAGCGATTGCGTTTC | GAAGAACTGGAGCGAACGAAG | 360 |
| Ginger92 | CGACTATTGTGCTTGGGTTGA | ACCATCGCCGTCGTACTAAA | 343 |
| Ginger97 | TTTATCCGGTTGGCTCAGC | GTATGTCTCTTTCAGCATTCCTCAC | 238 |

### 2.8. Statistical Analysis

The data of different parameters were assessed statistically by using one-way ANOVA in SPSS statistical software (version 23.0), and significant differences between the means (ex-

pressed as the mean value ± standard deviation) were identified using Duncan's multiple range test at $p < 0.05$.

## 3. Results

### 3.1. Growth Performance, Changes in the Plant Morphological Parameters, and Variation in the Absolute Growth Rate of the Disease-Free Plants during the Growth Period

At the beginning of planting, the plants grew very slowly with several light-green leaves, and most had only one thin tiller (Figure 1A). Thirty days later, the plant height increased rapidly and the leaves grew faster and became larger (Figure 1B). By 60 days after planting, most plants had produced 4–5 tender tillers with many green leaves (Figure 1C). Sixty days after planting, the plants continually grew vigorously and quickly, and the number of tillers and leaves was observed to increase greatly during until 120 days. After 120 days, the growth of aboveground parts of the plants slowed down gradually, and the leaf margins began to turn yellow (Figure 1D), as might be attributed to a progressive decrease in the atmospheric temperature.

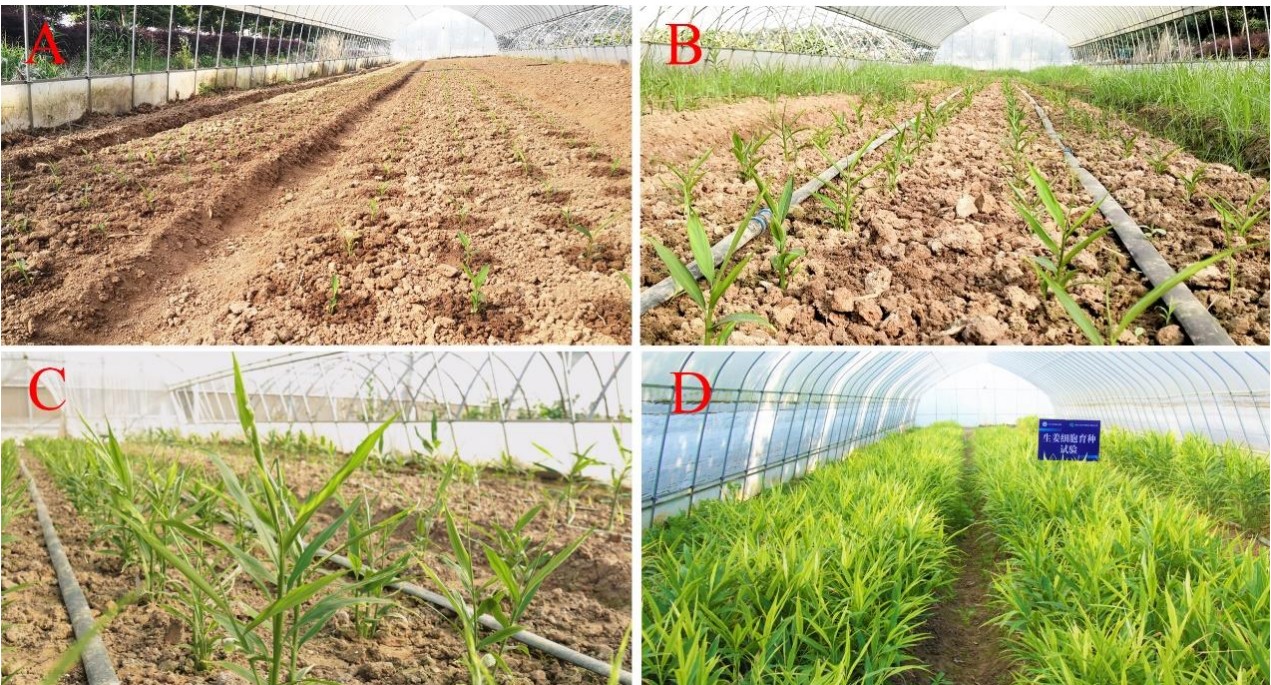

**Figure 1.** Field performance of the disease-free ginger plants cultivated in a simple plastic greenhouse under natural conditions. Photographs were taken 0 d (**A**), 30 d (**B**), 60 d (**C**) and 150 d (**D**) after planting, respectively.

Morphological indexes of the disease-free plants at different periods after planting in the field were analyzed as presented in Table 2. With the extension of the time, a continuously increasing trend was observed for the analyzed parameters including plant height, tiller number per plant, tiller diameter, leaf number per tiller, leaf length, and leaf width. Variance analysis showed that all the measured indexes exhibited a significant upward trend within 120 days after planting except leaf length/width ratio. During the growth period of 120–150 days, leaf number per tiller, leaf length, and leaf width did not change significantly, while there was still a significant increasing trend for the three indexes, namely plant height, tiller number per plant, and tiller diameter. For leaf length/width ratio, results showed that it was gradually increased after planting, and it reached the maximum (8.27) 120 days after planting. Thereafter, no significant change was found in leaf length/width ratio.

**Table 2.** Morphological indexes of the disease-free plants of ginger at different periods after planting.

| Days after Planting (d) | Plant Height (cm) | No. of Tillers per Plant | Tiller Diameter (mm) | No. of Leaves per Tiller | Leaf Length (cm) | Leaf Width (cm) | Leaf Length/Width Ratio |
|---|---|---|---|---|---|---|---|
| 0 | 8.54 ± 0.93 [f] | 1.12 ± 0.11 [f] | 2.61 ± 0.13 [f] | 5.54 ± 0.99 [e] | 5.28 ± 0.37 [e] | 1.06 ± 0.09 [e] | 4.99 ± 0.25 [d] |
| 30 | 15.72 ± 2.11 [e] | 2.06 ± 0.09 [e] | 4.54 ± 0.21 [e] | 6.42 ± 1.36 [d] | 8.38 ± 0.42 [d] | 1.40 ± 0.16 [d] | 6.04 ± 0.66 [c] |
| 60 | 31.80 ± 2.58 [d] | 4.16 ± 1.02 [d] | 6.51 ± 0.32 [d] | 7.76 ± 1.51 [c] | 11.40 ± 0.82 [c] | 1.76 ± 0.18 [c] | 6.50 ± 0.25 [c] |
| 90 | 52.46 ± 6.49 [c] | 10.72 ± 1.70 [c] | 7.56 ± 0.38 [c] | 9.78 ± 1.85 [b] | 17.42 ± 1.23 [b] | 2.36 ± 0.15 [b] | 7.39 ± 0.45 [b] |
| 120 | 73.23 ± 6.57 [b] | 12.34 ± 1.93 [b] | 10.90 ± 0.57 [b] | 14.36 ± 1.71 [a] | 25.86 ± 1.85 [a] | 3.12 ± 0.19 [a] | 8.29 ± 0.35 [a] |
| 150 | 80.01 ± 7.24 [a] | 13.40 ± 1.81 [a] | 11.52 ± 0.26 [a] | 14.84 ± 1.48 [a] | 27.22 ± 0.92 [a] | 3.30 ± 0.19 [a] | 8.27 ± 0.51 [a] |

Values (mean ± standard deviation) within columns followed by different superscript letters are significantly different according to Duncan's multiple range tests at the 5% level.

Changes in the absolute growth rate (AGR) of the morphological traits including plant height, tiller number per plant, tiller diameter, leaf number per tiller, leaf length, and leaf width at different periods after planting were shown in Figure 2. The AGR of both plant height and tiller number per plant reached the maximum during 60–90 days after planting, and showed a similar change trend during plant growth, i.e., a significant increase firstly and then a rapidly decrease (Figure 2A,B). Within 60 days after planting, the AGR of leaf number per tiller had no significant changes, while it was observed to have a significant increase trend during 60–120 days and then declined significantly (Figure 2C). The AGR of both leaf length (Figure 2D) and leaf width (Figure 2E) showed a similar change trend to the leaf number per tiller, indicating that the three traits related to ginger leaves had a similar growth behavior after planting. As for the AGR of tiller diameter, no significant difference was found between the first and the second 30-day intervals after planting (Figure 2F). Thereafter, it significantly declined during 60–90 days after planting, followed by a significantly higher rate during 90–120 days and a significantly lower value after 120 days (Figure 2F).

*3.2. Visually Screening, Morphological Characterization, and Main Yield Traits Evaluation of Somaclonal Variation at Harvest*

Visual screening of morphological variants was carried out during the growth of the disease-free plants. At the initial of the field experiment, the plants were almost uniform in morphological characteristics, and no obvious phenotypic differences were found among the plant populations. Approximately 90 days after planting, eight plants were observed unexpectedly to exhibit clear differences in morphological characteristics, such as tiller number, leaf size, leaf color, and plant height. These plants were distributed in different regions of the field. Subsequently, these variants were under continuous close observation till harvest, and the observed morphological differences were stable over time. At harvest, the normal-type plants had erect tillers with long and narrow green leaves (Figure 3A). Based on observed differences in tiller number, plant height, leaf color, and leaf size, the eight plants were largely classified into two variation types (VT), and three of them were classified as VT1 and the remaining were assigned to VT2 (Figure 3; Tables 3 and 4).

As shown in Figure 3 and Table 3, clear differences were observed in the morphological characteristics of the three types of plants. Among the field-grown plants, the VT1 had the shortest plant height (72.95 cm), the most tillers (27.50 per plant), and the smallest leaves (23.70 cm in length and 2.48 cm in width) with yellow-green color (Figure 3B; Table 3). Nevertheless, the VT2 had a higher plant height (89.40 cm) whereas developed less (10.50 per plant) but thicker tillers (11.63 cm) with larger (27.48 cm in length and 2.48 cm in width) and dark-green leaves compared with the other two types of plants (Figure 3C; Table 3). These plants differed significantly in the leaf relative chlorophyll content expressed as SPAD value, with a descending order of the VT2 > the normal type > the VT1 (Table 3), which was consistent with their observed leaf color. Both the normal-type and the VT1 were characterized by significantly reduced leaf length and leaf width compared with the VT2. As for leaf length/width ratio, a significant difference was found between the two variation types, while both of them had no significant differences from the normal-type

plants (Table 3). Briefly, the VT2 plants were characterized by significantly increased leaf SPAD value, plant height, tiller diameter, and leaf size, and significantly reduced tiller number compared with the other two types. There were no significant differences between the VT1 and the normal type in all the analyzed parameters except leaf SPAD value, tiller number, and leaf width.

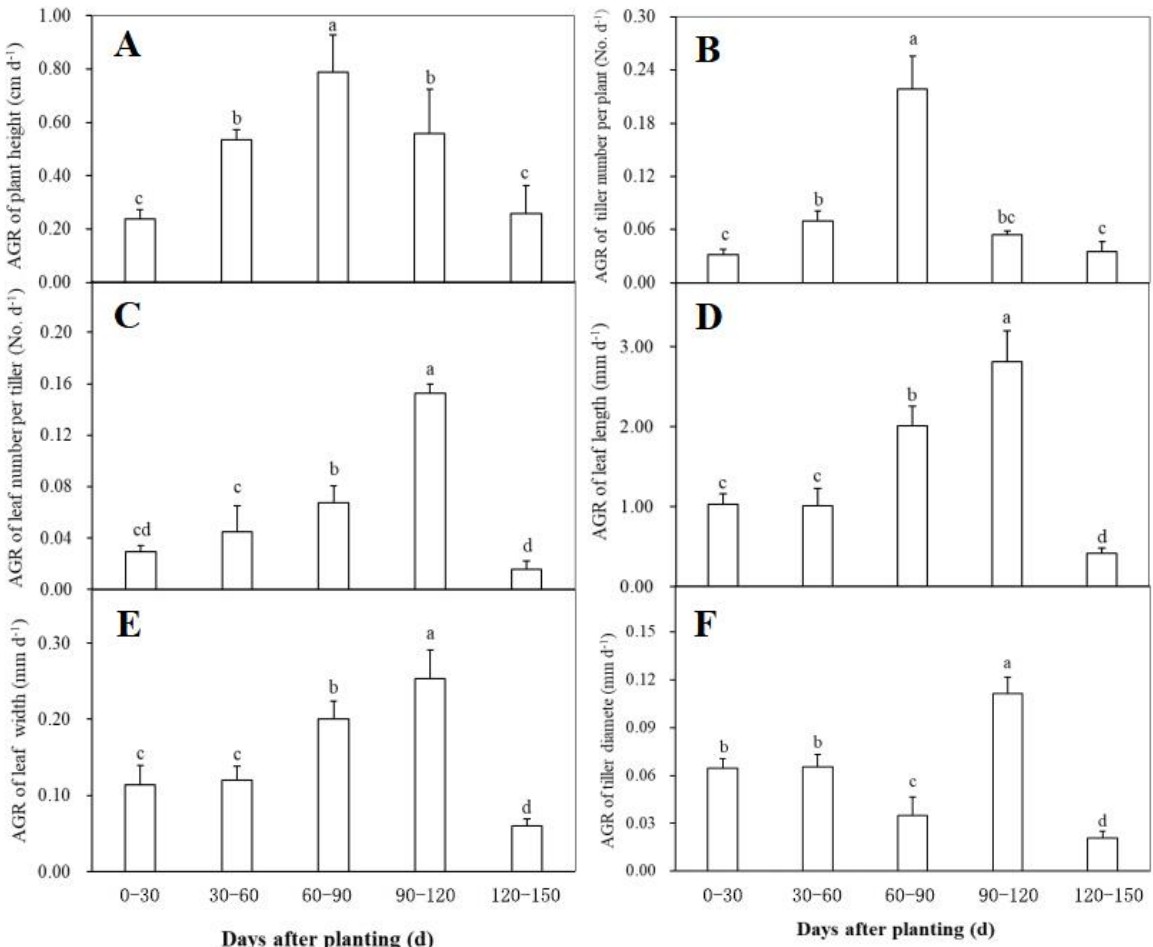

**Figure 2.** The relative growth rate (RGA) of plant height (**A**), tiller number per plant (**B**), leaf number per tiller (**C**), leaf length (**D**), leaf width (**E**), and tiller diameter (**F**) of the disease-free ginger plants grown in a simplified plastic greenhouse at different periods after planting. Means ± standard deviation (SD) with different letters indicate significantly difference at $p < 0.05$.

In addition, the rhizome features of the three types were also different from each other. As shown in Figure 3D and Table 4, the normal-type plants produced medium-sized rhizomes (19.90 cm in length and 4.50 cm in height) with medium-number knobs (26.50 per plant) compared with the two variation types. The rhizomes of the VT1 were observed to arrange in multiple lays, with the largest number (32.25 per plant) but the smallest size of knobs and the most fibrous roots (Figure 3E and Table 4). The rhizomes of the VT2 arranged in two rows, with the largest size (20.63 cm in length and 4.95 cm in height) whereas the least number of rhizome knobs (16.50 per plant) (Figure 3F and Table 4). No differences were observed in the color of the rhizome among the three types when harvesting, and the outer skin of all the rhizomes was yellowish and the bases of all the aerial shoots appeared in pink. As to the rhizome fresh weight per plant, the VT2 reached the largest (269.48 g), followed by the VT1 (170.83) and the normal type (147.80). No significant differences were found in the analyzed main yield traits between the VT1 and the normal type, and it was also observed between the VT2 and the normal plants except rhizome fresh weight per

plant. Nevertheless, there were significantly different between the two variation types in all the measured yield traits.

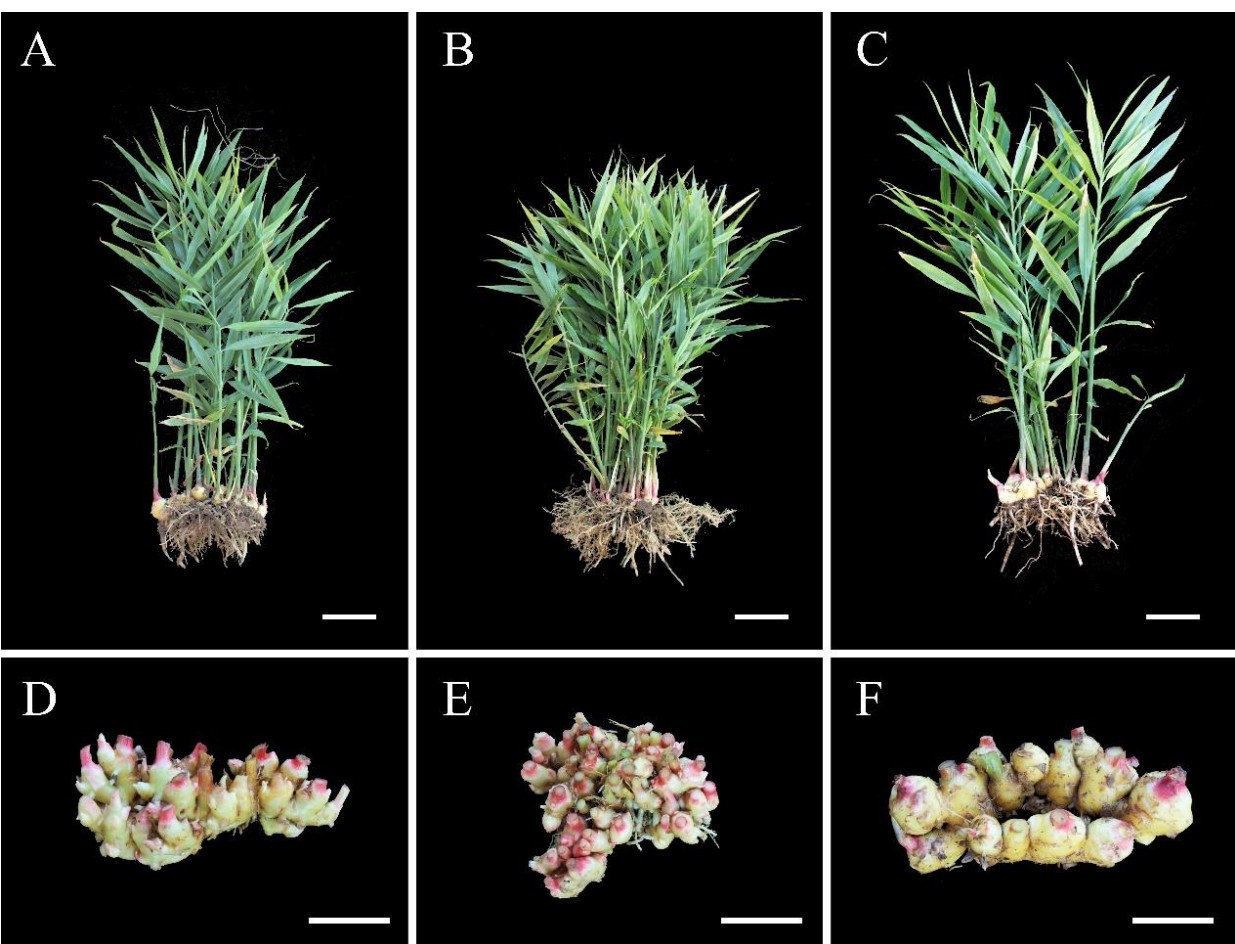

**Figure 3.** Morphological features of the aerial shoots and rhizomes of the three types of ginger plants at harvest. (**A,D**) the normal-type plant; (**B,E**) the VT1; (**C,F**) the VT2. Scale bars = 10 cm.

**Table 3.** Main morphological parameters of the normal-type ginger and the two variation types at harvest.

| Plant Types | Leaf SPAD Value | Plant Height (cm) | Tiller Diameter (mm) | No. of Tillers per Plant | Leaf Length (cm) | Leaf Width (cm) | Leaf Length/Width Ratio |
|---|---|---|---|---|---|---|---|
| Normal type | 48.70 ± 0.85 [b] | 80.73 ± 4.29 [b] | 7.65 ± 0.64 [b] | 14.67 ± 1.15 [b] | 24.55 ± 0.49 [b] | 2.80 ± 0.14 [b] | 8.78 ± 0.62 [ab] |
| VT1 | 44.53 ± 0.88 [c] | 72.95 ± 7.28 [b] | 6.23 ± 0.67 [b] | 27.50 ± 4.95 [a] | 23.70 ± 1.60 [b] | 2.48 ± 0.13 [c] | 9.60 ± 0.89 [a] |
| VT2 | 54.88 ± 1.65 [a] | 89.40 ± 4.93 [a] | 11.63 ± 1.65 [a] | 10.50 ± 1.22 [c] | 27.48 ± 1.19 [a] | 3.33 ± 0.17 [a] | 8.27 ± 0.41 [b] |

Different lowercase letters in the same column represent significant difference at $p < 0.05$ according to Duncan's range test. VT: variation type.

**Table 4.** Main yield traits of the normal-type ginger and the two variation types at harvest.

| Plant Types | Rhizome Length (cm) | Rhizome Height (cm) | No. of Rhizome Knobs per Plant | Rhizome Fresh Weight per Plant (g) |
|---|---|---|---|---|
| Normal type | 19.90 ± 4.81 [ab] | 4.50 ± 0.16 [ab] | 26.50 ± 2.12 [ab] | 147.80 ± 22.63 [b] |
| VT1 | 15.23 ± 2.89 [b] | 4.05 ± 0.21 [b] | 32.25 ± 6.40 [a] | 170.83 ± 14.91 [b] |
| VT2 | 20.63 ± 4.05 [a] | 4.95 ± 0.42 [a] | 16.50 ± 4.43 [b] | 269.48 ± 35.47 [a] |

Means with the same letters within columns were not significantly different at $p < 0.05$ by Duncan's range test. VT: variation type.

### 3.3. Evaluation of Stomatal Characteristics, Ploidy Level, and Chromosome Number of Somaclonal Variation

Variations of the stomatal characteristics of the three types were presented in Table 5. The VT1 had the least aperture length and width, and guard cell length and width among the three types of plants, while no significant differences were found between the VT1 and the normal type in in the four parameters except aperture width. The VT2 plants had significantly larger aperture length, aperture width, and guard cell length compared with the other two groups, while it did not differ significantly among the three types for aperture width. The highest stomatal density was recorded in the normal type, as was significantly higher than that of the VT2, while it did not differ significantly from the VT1. Briefly, the VT2 had significantly larger stomatal apertures and guard cell length whereas significantly lower stomatal density among the three plant types, while no significant differences were observed between the normal type and the VT1 in all the measured parameters except aperture width.

**Table 5.** Differences in stomatal parameters among the normal-type ginger and the two variation types at harvest.

| Plant Types | Aperture Length (μm) | Aperture Width (μm) | Guard Cell Length (μm) | Guard Cell Width (μm) | Stomatal Density (No./mm²) |
|---|---|---|---|---|---|
| Normal type | 26.83 ± 2.46 [b] | 13.23 ± 2.09 [b] | 38.56 ± 2.07 [b] | 23.60 ± 1.41 [a] | 85.04 ± 6.21 [a] |
| VT1 | 25.44 ± 2.51 [b] | 11.88 ± 2.19 [c] | 37.68 ± 1.86 [b] | 23.72 ± 1.75 [a] | 81.34 ± 5.95 [a] |
| VT2 | 29.3 ± 2.66 [a] | 15.68 ± 2.91 [a] | 42.01 ± 2.17 [a] | 24.67 ± 2.14 [a] | 65.10 ± 4.77 [b] |

Means within (± standard deviation) within a column followed by different letters are significantly different at the 5% level according to Duncan's multiple range test. VT: variation type.

Results of flow cytometry analysis of the three types of plants were shown in Figure 4. Both the normal-type plants (Figure 4A) and the VT1 (Figure 4B) had only one peak situating at a value of approximately 180 in the histogram, which suggested that they were not mixoploids or chimeras. All the plants of the VT2 showed two dominant $G_1$ peaks of relative fluorescence intensity at 180 and 360 (Figure 4C), indicating their mixoploidy.

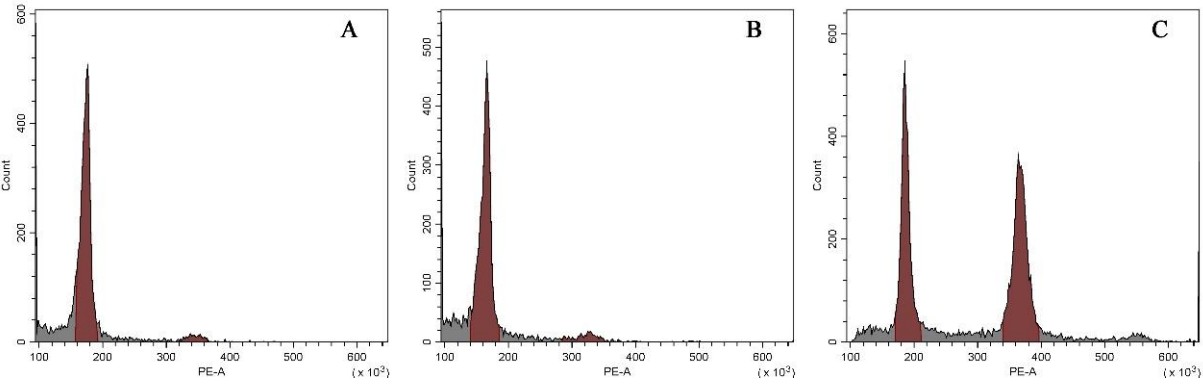

**Figure 4.** Histograms of relative 2C DNA content in the nuclei isolated from the normal-type ginger (**A**) and the two morphological variation types, namely the VT1 (**B**) and the VT2 (**C**).

To confirm the results of flow cytometry analysis, chromosome number counting was also conducted as shown in Figure 5. Both the normal-type plants (Figure 5A) and the three plants of the VT1 (Figure 5B) were found to have 22 chromosomes. Nevertheless, it was observed that the root-tip cells with 44 (Figure 5C) and 22 chromosomes (Figure 5D) co-existed in all the plants of the VT2, which was consistent with the results of their DNA content analysis. Previous studies showed that the basic chromosome number of ginger is $x = 11$ [37]. Therefore, the results of flow cytometry analysis and chromosome counting in this study confirmed that the normal-type plants of 'Wuling' ginger and the VT1 were

both diploid ($2n = 2x = 22$), while the VT2 was mixoploid with diploid and tetraploid cells ($2n = 4x = 44$).

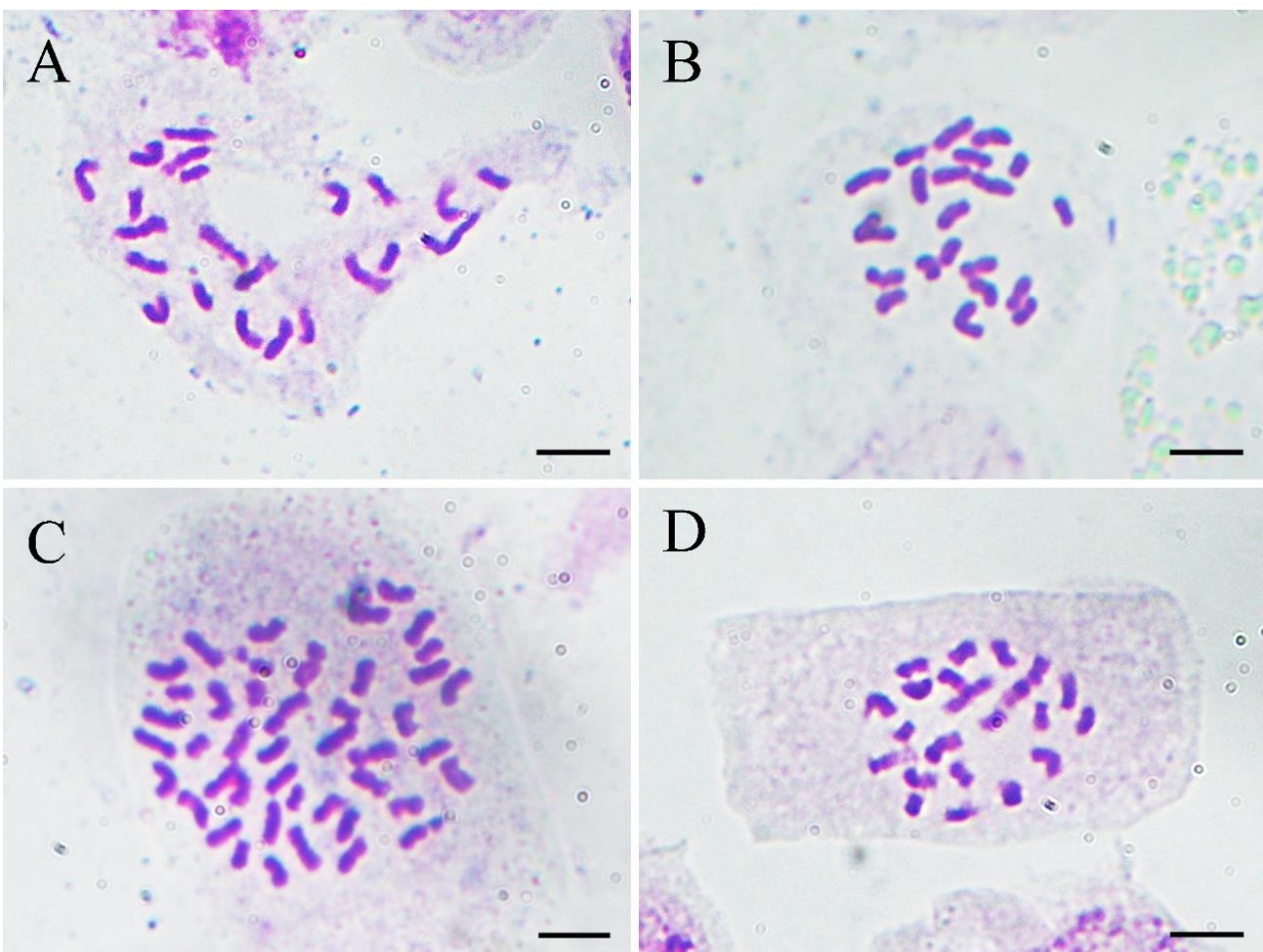

**Figure 5.** Micrographs ($\times 1000$) of somatic chromosomes in the root tips of the normal-type ginger (**A**), the VT1 (**B**), and the VT2 (**C,D**). Scale bars = 10 μm.

### 3.4. SSR Characterization of the Two Variation Types

Out of the 20 pairs of SSR primers adopted, only 11 markers including Ginger58 (Figure 6A), Ginger92 (Figure 6B), and Ginger77 (Figure 6C) produced reproducible and clear DNA bands on 8% non-denaturing acrylamide gels. However, only one pair of primers, namely Ginger77, detected banding pattern changes within the ten plants from the three groups (Figure 6C). In detail, one plant of the VT1 and all the five plants of the VT2 showed one new allele with primer Ginger77 (Figure 6C), indicating that genomic variation occurred in these plants.

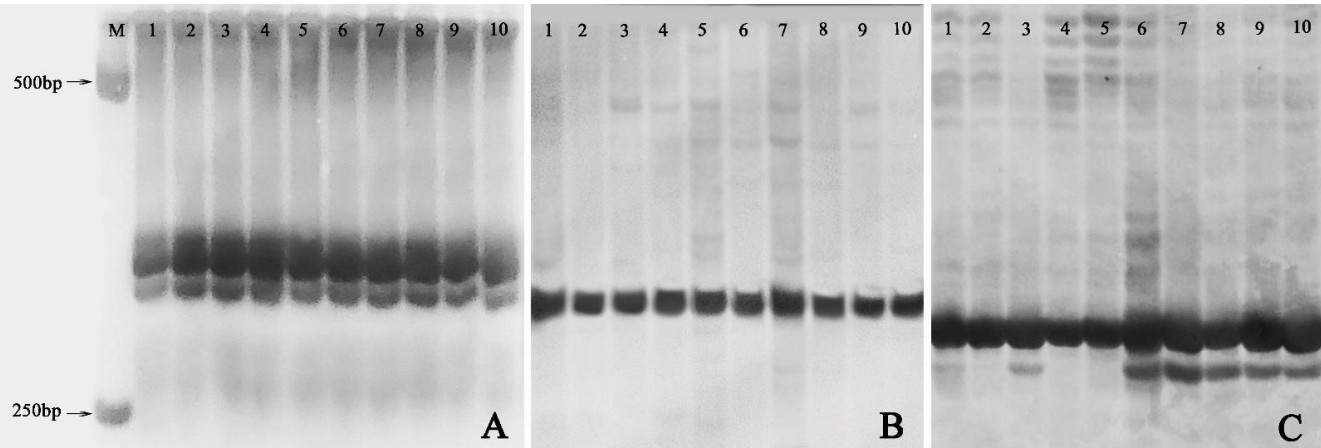

**Figure 6.** SSR profiles of the normal-type plants, the VT1, and the VT2. (**A**) Banding patterns of SSR marker Ginger 58; (**B**) Banding patterns of SSR marker Ginger 92; (**C**) Banding patterns of SSR marker Ginger77. M: 1000 bp DNA marker; 1–2: randomly selected two plants of normal-type plants; 3–5: the VT1; 6–10: the VT2.

## 4. Discussion

The production of disease-free ginger rhizomes in a high-efficient and low-cost way is very necessary for this economically important crop. Plant tissue culture provides an efficient and feasible approach for the large-scale production of disease-free ginger plants [16–18], while it will take at least 2 to 3 years to produce seed rhizomes that can be used commercially [21–23]. Plant morphological and growth properties are closely involved in the rhizome yield, which can provide useful information for field management of ginger [22–24]. Therefore, the field performance of the disease-free plants of 'Wuling' ginger was evaluated during the whole growing season in this study. In addition, somaclonal variants induced in tissue culture are important sources of genetic variability for crop improvement [26,27]. Therefore, characterization of somaclonal variants was also conducted in this study by agronomic, cytological, and molecular evaluation, with the aim to screen potential plant lines for future breeding in ginger.

Generally, the growth periods of conventionally propagated ginger plants can be divided into the germinating stage, seedling stage, flourishing growing stage, and rhizome dormant stage, and both the underground parts and the rhizomes develop quickly in the flourishing growing stage [38]. However, few studies have been conducted in detail on the growth habits of the in vitro regenerated plants in ginger [23]. To understand the field performance of the in vitro-produced disease-free ginger plants, continuous observation and measurement of the morphological characteristics were performed at 30-day intervals during the whole growing season in this study. These results indicated that the period of 60–120 days after planting (from late July to late October) was the most active growth stage for the disease-free plants, which was similar to the growth properties of the conventional plants in the flourishing growing stage. For the tissue-cultured ginger plants, Ren et al. (2020) [23] found that both plant height and tiller diameter increased rapidly in the early stage and slowed down in the late stage, which was consistent with our results. Corresponding agricultural management practices such as irrigation, fertilizer application, and shading should be enhanced in the most active growth stage for the in vitro-propagated disease-free plants in ginger.

Somaclonal variation has been observed in the process of plant tissue culture, and the frequency is relatively high when cultures are maintained in vitro for a long time [25,26]. In ginger, previous studies also observed somaclonal variation in tissue culture-derived plant populations [32–34]. In this study, eight plants were screened as somaclonal variants by close observation amongst approximately 2000 plants maintained in vitro for three years. The frequency of somaclonal variation in ginger was approximately 0.4%, which was ex-

tremely lower compared with other plant species such as caladium (*Caladium × hortulanum* Birdsey) [27]. Most ginger cultivars differ in the quantitative traits such as plant height, tiller number, and rhizome size, which were easily influenced by growth environmental conditions, while only subtle differences are observed in quality traits such as leaf color and leaf shape [24,38]. Therefore, it is relatively difficult to distinguish somaclonal variants among a large number of regenerants based on morphological observation, which may cause some somaclonal variants are escaped from screening at an early stage. This could explain the reason for the low frequency of somaclonal variation in this study in a certain extent. It is reported that the frequency of somaclonal variation is determined by many factors such as genotype [26]. Therefore, the low frequency of variation might also be attributed to high genetic stability of ginger during in vitro culture.

Ginger is generally propagated by underground rhizomes. Therefore, the yield and knob size of the rhizomes have a direct impact on their reproductive coefficient and economic benefit. However, in the first generation of in vitro cultured ginger plants, the productivity was found to be much lower and the rhizome knobs were smaller compared with rhizome-grown plants [21–23]. In our study, we also found that the rhizome fresh weight per plant was very low for the three types of plants (Table 4), and the rhizome knob size was also too small to be used as seed rhizomes, especially for the normal-type plants and the VT1 (Figure 3A,B). This might be due to the fact that the disease-free ginger plants have no extra nutrition supply from the mother rhizomes in the first growing season. Previous studies suggested that the ginger yield had a significantly positive correlation with tiller diameter and plant height, whereas was negatively correlated with tiller number [22–24]. Therefore, a significantly higher rhizome fresh weight of the VT2 plants could be attributed to their significantly increased plant height and tiller diameter, and significantly reduced tiller number.

In addition to morphological and agronomic characterization, stomatal characteristics, flow cytometry analysis, and chromosome counting were also used to confirm the ploidy level of the morphological variation plants in this study. Changes in chromosome number have been found in somaclonal variation, such as aneuploidy [27], polyploidy [35], and mixoploidy [39]. Our results showed that both the normal type (Figure 5A) and the VT1 (Figure 5B) were diploid with $2n = 22$ chromosomes, while the VT2 was mixoploid, consisting of diploid and tetraploid cells (Figure 5C,D). Ploidy level has profound effects on the morphological, cellular, and physio-biochemical characteristics in plants [37,40]. Wang et al. [37] found that diploid ginger cultivars in China had stronger but fewer tillers and bigger rhizomes with larger size of knobs, while the mixoploid cultivars that occurred naturally were just the opposite. In our study, the VT2 showed increased leaf SPAD value, plant height, tiller number, and rhizome and leaf size, and reduced tiller number and leaf length/width compared with the diploid normal type and the VT1, which was consistent with the previous study [37]. In addition, generally stomatal features including guard cell length/aperture length and stomatal density have been found to be closely related to the plant ploidy level [27,37,39]. In this study, significantly larger guard cells/apertures and significantly lower stomatal density were also observed in the VT2, a mixoploid.

DNA-based molecular markers have been widely used in analyzing the genetic stability of in vitro regenerated plants [26,27,34]. In this study, one pair of SSR primers (Ginger77) detected the polymorphisms among the three types of plants, and six out of eight morphological variants showed one new allele of similar size (Figure 6C), thus confirming the genetic instability of the in vitro regenerated plants at the molecular level. Although the VT1 showed stable and clear differences from the normal type, no polymorphisms were observed between the remaining two plants of the VT1 and the normal-type plants. These results indicated that SSRs could not effectively detect all genetic variants derived from tissue culture in ginger, which genome size is approximately 1.59 Gb with 3.6% heterozygosity [41]. Ioannidis et al. [42] also indicated that SSRs were not effective in detecting all clonal variability originated from all possible random mutations in Cannabis due to its relatively big genome size. Further, plant phenotypic variation is determined not only by

genetic factors but also by epigenetic causes [25]. Therefore, DNA methylation variation and more SSR primers should be analyzed to present more molecular information about the two variation types in the future.

## 5. Conclusions

This study comprehensively analyzed the field performance and somaclonal variation of the disease-free plants of 'Wuling' ginger produced by in vitro culture. By continuous observation and measurement, the plants were found to grow vigorously within 0–120 days after planting, and the most active growth was recorded during 60–120 days. A relatively low frequency of somaclonal variation was also observed among the field-grown plants, as was further characterized by agronomic, cytological, and molecular analysis. Significant differences in morphological, agronomic, and stomatal parameters were found among the three types of plants, as might be highly related to the changes in their ploidy level. DNA band profile changes were also detected in some somaclonal variants by SSR analysis. The findings of this study might contribute to the commercial production of disease-free seed rhizomes in ginger, and the screened somaclonal variants including diploid and mixoploid lines could provide valuable germplasm resources for future ginger breeding.

**Author Contributions:** Conceptualization, X.Z., S.Y. and X.C.; data curation, X.Z., Y.Z. (Yiming Zhang) and Y.W.; formal analysis, X.Z., Q.J. and Y.Z. (Yongxing Zhu); funding acquisition, X.C., J.Y. and Y.L.; investigation, X.Z., D.J. and L.H.; methodology, X.Z. and S.Y.; resources, X.C., J.Y. and Y.L.; writing—original draft, X.Z. and S.Y.; writing—review and editing, X.C. All authors have read and agreed to the published version of the manuscript.

**Funding:** This research was funded by the Key Research and Development Project of Hubei Province, under grant no. 2021BBA096 and no. 2022BBA0061.

**Data Availability Statement:** The data presented in this study are available on request from the corresponding authors.

**Conflicts of Interest:** The authors declare no conflict of interest.

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
