# Peer review of "Field Performance of Disease-Free Plants of Ginger Produced by Tissue Culture and Agronomic, Cytological, and Molecular Characterization of the Morphological Variants"

_agronomy, doi:10.3390/agronomy13010074_

Round 1
Reviewer 1 Report
1. The authors constantly and excessively use the concept of disease-free plants, but there is no evidence for this.
2. Is ‘Wuling’ synthetic cultivar or clonal cultivar? There is no data in the method which source material was used.
3. An in vitro plant propagation protocol has not been described.
4. It is recommended to combine the sections: Morphological measurements" and "Visual screening of morphological variants and main agronomic traits analysis".
5. It is recommended to combine the sections "Growth performance of the disease-free plants, Changes in the plant morphological parameters during the growth period, Changes in the absolute growth rate of the plants".
6. The section "Growth performance of the disease-free plants" is overly detailed. In figure 1, for example, you can leave only pictures 1A, B, C, F.
7. To identify of somaclonal variation, SSR analysis should preferably be done at the in vitro stage, and not ex vitro.
8. The authors claim that they have determined relative nuclear DNA content. However, they determined the level of ploidy by flow cytometry. Since the standard was not used in determining the relative nuclear DNA content.
9. In the "Screening of somaclonal variation" section, the terms "medium-sized, largest number, smallest size et al. should have numeric values. Probably the "Screening of somaclonal variation" section should be combined with section the "Identification of the aboveground morphological characteristics at harvest" .
10. In keywords add "SSR-analysis".
Author Response
Point 1: The authors constantly and excessively use the concept of disease-free plants, but there is no evidence for this.
Response 1:
Thank you for your helpful comments. In fact, in vitro production of disease-free ginger plants is a routine work in our lab. The disease-free nature of these plants was identified by RT-PCR using specific primers of several soil-borne diseases in ginger, such as bacterial wilt disease and rhizome rot. We have revised the sentence in “Materials and methods”: In our previous work, tissue-cultured plants of ‘Wuling’ ginger were obtained by shoot tip culture and were proved to be disease-free by RT-PCR test (data not published).
Point 2: Is ’Wuling’ synthetic cultivar or clonal cultivar? There is no data in the method which source material was used.
Response 2:
As mentioned in the “introduction”, conventional sexual hybridization is limited in ginger for its poor flowering, pollen viability, and seed set, resulting in its narrow genetic variability. Presently systematic breeding is the most frequently used in ginger. Wuling’ ginger is a well-known local ginger cultivar in Wuling Mountain area, China. We have introduced it in the method.
Point 3: An in vitro plant propagation protocol has not been described.
Response 3:
We have added a sentence in the revised manuscript: For mass production of the plants to ensure adequate materials for further experiment, the maintained plants were cut under a biological safety cabinet into shoot clusters of about 1 cm in height, and cultured on the above medium at 25 ± 1 °C and 14 h/10 h (light/dark) cycles under cool white fluorescent lamps of about 55 μmol m−2 s−1.
Point 4: It is recommended to combine the sections: Morphological measurements" and "Visual screening of morphological variants and main agronomic traits analysis".
Response 4:
Thank you for your kind and good suggestion. We have combined the two sections in the revised paper.
Point 5: It is recommended to combine the sections "Growth performance of the disease-free plants, Changes in the plant morphological parameters during the growth period, Changes in the absolute growth rate of the plants".
Response 5:
Thank you for your good suggestion. We have combined the three sections in the revised paper.
Point 6: The section "Growth performance of the disease-free plants" is overly detailed. In figure 1, for example, you can leave only pictures 1A, B, C, F.
Response 6:
We have deleted two pictures according to your kind and good suggestion, and the corresponding sentences were also revised in the results in the revised paper.
Point 7: To identify of somaclonal variation, SSR analysis should preferably be done at the in vitro stage, and not ex vitro.
Response 7:
Yes, we are quite agreeing with you. However, we found the fact that a very low frequency of somaclonal variation occurred in tissue-culture derived plants in ginger according our work and previous reports (Mohanty et al., Genetic Stability of Micropropagated Ginger Derived from Axillary Bud through Cytophotometric and RAPD Analysis, 2008; Jain et al., Effect of carbendazim on in vitro conservation and genetic stability assessment in Curcuma longa. and Zingiber officinale, 2018; George et al., Direct in vitro regeneration of medicinally important Indian and exotic red‑colored ginger (Zingiber officinale Rosc.) and genetic fidelity assessment using ISSR and SSR markers, 2022 ). Therefore, morphological variation plants were firstly screened in the field and then subjected to SSR analysis in this study, with an aim to increase the efficiency of identifying somaclonal variation.
Point 8: The authors claim that they have determined relative nuclear DNA content. However, they determined the level of ploidy by flow cytometry. Since the standard was not used in determining the relative nuclear DNA content.
Response 8:
Thank you for your comments. The standard is needed in determining the absolute DNA content of plant species. In our study, the relative nuclear DNA content was determined by FCM. As shown in Fig. 4A and Fig. 4B, only one peak situating at a value of about 180 in the histogram, indicating they have a similar ploidy level. The plants of the VT2 showed two dominant G1 peaks of relative fluorescence intensity at about channel both 180 and 360 (Fig. 4C), indicating their mixoploidy. Subsequent chromosome counting may confirm the results of flow cytometry analysis.
Point 9: In the "Screening of somaclonal variation" section, the terms "medium-sized, largest number, smallest size et al. should have numeric values. Probably the "Screening of somaclonal variation" section should be combined with section the "Identification of the aboveground morphological characteristics at harvest" .
Response 9:
We have made a major revision in this part according to your helpful suggestions. numeric values were added to corresponding positions, and three part including "Screening of somaclonal variation" section , "Identification of the aboveground morphological characteristics at harvest" and “evaluation of main yield traits at harvest” were combined.
Point 10: In keywords add "SSR-analysis"
Response 10:
"SSR-analysis" has been added in keywords.

Reviewer 2 Report
Dear Authors,
The manuscript entitled "Field performance of disease-free plants of ginger produced by tissue culture and agronomic, cytological, and molecular characterization of the morphological variants" by Zhao et al describes the field performance of in vitro cultured Zingiber officinale plants and tries to establish the performance and genetic stability of the cultured plants. The manuscript must be improved in several sections, by addind relevant information, before considering the acceptance. See the attached file.
Yours sincerely

Author Response
Response to Reviewer 2 Comments
Point 1: Were the plants checked for being disease-free? If yes, how?
Response 1: Thank you for your helpful comments. In fact, in vitro production of disease-free ginger plants is a routine work in our lab. The disease-free nature of these plants was identified by RT-PCR using specific primers of several soil-borne diseases in ginger, such as bacterial wilt disease and rhizome rot. We have revised the sentence in “Materials and methods”: In our previous work, tissue-cultured plants of ‘Wuling’ ginger were obtained by shoot tip culture and were proved to be disease-free by RT-PCR test (data not published).
Point 2: Please add the photoperiod and the photosynthetic photon flux density.
Response 2: We have added the photoperiod 28/20℃(14 h/10 h) and the light intensity level between 100 and 130 lmol m-2 s-1.
Point 3: Add some information about climatic conditions (air temperature, humidity etc.).
Response 3: We have added: The annual average temperature of the site ranges from 15.9 to 16.5℃, and the annual mean rainfall is about 1100-1150 mm.
Point 4: Checked for what? Once you used NanoDrop you can check both purity and concentration ....
Response 4: We corrected this sentence: Quantity and purity of the isolated DNA were checked by electrophoresis with 0.8% agarose gel at a constant voltage of 100 V for 30 min and a NanoDrop One C spectrophotometer.
Point 5: Please add the gradually decreasing annealing temperature.
Response 5: We have added the gradually decreasing annealing temperature: primer firstly annealing at 63°C for 30 s with 1ºC reduction per cycle,. Thank you for your careful review and helpful suggestions.
Point 6: Was there any experimental design (any blocking) in order to assess there was no environmental influence?
Response 6: Thank you for your constructive comments. We have realized that this sentence was imprecisely, and have revised it: Subsequently, these variants were under continuous close observation till harvest, and the observed morphological differences were stable over time.
Point 7: Are there any information about SSR primers generated amplicons in the population used for the in vitro culture e.g. the number of alleles and their range for each locus, alleles’ frequencies, the mean of the different alleles per locus, the effective alleles, the mean observed and the mean expected heterozygosity, the percentage of polymorphic loci, Nei genetic distance and Nei genetic identity, the polymorphism information content etc.???
Response 7: At present, we cannot provide information mentioned. In fact, ginger cultivars have a narrow genetic background for their difficulty in sexual hybridization. We will develop more specific SSR primers for a more comprehensive analysis of genetic stability in the future.
Point 8 and 10: SSRs can effectively detect all clonal variability originated by all possible random mutations that occurred in Zingiber officinale genome size 1.59 Gb???
Response 8: In my opinion, at present SSRs is not very effective in detecting clonal variability originated by all possible random mutations that occurred in Zingiber officinale. In fact, we found that a very low frequency of somaclonal variation occurred in tissue-culture derived plants in ginger according our work and previous reports by SSR or other molecular markers analysis (Mohanty et al., Genetic Stability of Micropropagated Ginger Derived from Axillary Bud through Cytophotometric and RAPD Analysis, 2008; Jain et al., Effect of carbendazim on in vitro conservation and genetic stability assessment in Curcuma longa. and Zingiber officinale, 2018; George et al., Direct in vitro regeneration of medicinally important Indian and exotic red‑colored ginger (Zingiber officinale Rosc.) and genetic fidelity assessment using ISSR and SSR markers, 2022 ). The SSR primers or other molecular markers need more studies in ginger.
Point 9: It is more part of the results.
Response 9: We have deleted this part.
Point 11: Are the 11 primer pairs very limited for evaluating the somaclonal variation or the genetic homogeneity???
Response 11: Yes, we agree with you that it is limited for evaluating the somaclonal variation or the genetic homogeneity. Studies about SSR primers in ginger is limited now, and more efforts should be made.
Point 12: Is this one pair of markers capable to detect polymorphisms, confirming somaclonal variation??? Are the SSR markers amplifying coding regions? If yes, can one locus influence quantitative traits such as plant height, tiller number, and rhizome size, which were easily influenced by growth environmental conditions as you typically mentioned in lines 395-397???
Response 12: 1) One pair of markers was not enough to confirm somaclonal variation, and we corrected it : thus confirming the genetic stability of the in vitro regenerated plants of ginger at the molecular level.
2) Somaclonal variation including genetic and epigenetic alterations can occur spontaneously in a variety of plant species during tissue culture, as can be ascertained by morphological, cytological, biochemical, and molecular methods. In our study, significant differences in morphological, agronomic, stomatal parameters, and chromosome number were found among the three types of plants, and DNA band profile changes were also detected in some morphological variants by SSR analysis. Therefore, these comprehensive factors resulted in diverse morphological changes. Of course, more detail molecular mechanisms like DNA methylation variation should be analyzed to reveal the epigenetic factors of the morphological variants in ginger in the future.

Round 2
Reviewer 1 Report
The manuscript has been sufficiently improved
Author Response
Thank you very much for your helpful and kindful suggestions.
Reviewer 2 Report
Dear Authors,
Many changes according to the comments were introduced.
Nevertheless, you did not mention that SSRs, in general, could not effectively detect all clonal variability originated by all possible random mutations that occurred in Zingiber officinale, which genome size is about 1.59 Gb. Moreover, as you dealing with quantitative traits such as plant height, tiller number, and rhizome size, which are easily influenced by growth environmental conditions as you mentioned, you do not present more information about molecular characterization of the two variation.
You characteristically state (agronomy-2024734-peer-review-v2.pdf file) in line 396-397 that the results are “…indicating that genomic variation occurred in these plants.” However, in line 484-486 they state as a conclusion that the results are “….confirming the genetic stability of the in vitro regenerated plants of ginger at the molecular level.”, although “…. one pair of SSR primers (Ginger77) detected the polymorphisms among the three types of plants, and six out of eight morphological variants showed one new allele of similar size.” (Lines 482-484). These statements are contradictory and a bit confusing.
Please rephrase.
Author Response
- Nevertheless, you did not mention that SSRs, in general, could not effectively detect all clonal variability originated by all possible random mutations that occurred in Zingiber officinale, which genome size is about 1.59 Gb. Moreover, as you dealing with quantitative traits such as plant height, tiller number, and rhizome size, which are easily influenced by growth environmental conditions as you mentioned, you do not present more information about molecular characterization of the two variation. Reply:Thank you helpful suggestions . We added two references and revised these sentences: "
These results indicated that SSRs could not effectively detect all genetic variants derived from tissue culture in ginger, which genome size is approximately 1.59 Gb with 3.6% heterozygosity [41]. Ioannidis et al. [42] also indicated that SSRs were not effective in detecting all clonal variability originated from all possible random mutations in Cannabis due to its relatively big genome size. Besides, plant phenotypic variation is determined not only by genetic factors but also by epigenetic causes [25]. Therefore, DNA methylation variation and more SSR primers should be analyzed to present more molecular information about the two variation in the future."
-
"These statements are contradictory and a bit confusing. Please rephrase. "Reply, we have rephased and corrected these sentences.